# Effect of Yogurt Fermented by *Lactobacillus Fermentum* TSI and *L. Fermentum* S2 Derived from a Mongolian Traditional Dairy Product on Rats with High-Fat-Diet-Induced Obesity

**DOI:** 10.3390/foods9050594

**Published:** 2020-05-06

**Authors:** Won-Young Cho, Go-Eun Hong, Ha-Jung Lee, Su-Jung Yeon, Hyun-Dong Paik, Yoshinao Z. Hosaka, Chi-Ho Lee

**Affiliations:** 1Department of Food Science and Biotechnology of Animal Resources, Konkuk University, Seoul 05029, Korea; ready1838@naver.com (W.-Y.C.); lhjj@konkuk.ac.kr (H.-J.L.); hdpaik@konkuk.ac.kr (H.-D.P.); 2Lab supervisor, Factory, Royal Canin Korea, Jeollabuk-do 54325, Korea; vhdrh1004@naver.com; 3Department of Food Science and Biotechnology, Kangwon Institute of Inclusive Technology (KIIT), Kangwon National University, Gangwon-do 24341, Korea; sujung0811@gmail.com; 4Faculty of Agriculture, Joint Department of Veterinary medicine, Tottori University, Tottori 680-8550, Japan; y-hosa@muses.tottori-u.ac.jp

**Keywords:** yogurt, mongolian fermented dairy product, lactic acid bacteria, cholesterol, adiponectin, adipocyte

## Abstract

This study aimed to investigate the metabolic effect of yogurt fermented by *Lactobacillus fermentum* TSI and S2 isolated from a Mongolian traditional dairy product on rats with high-fat-diet-induced obesity. Quality characteristics of yogurt fermented by commercial starter (CON), *L. fermentum* TSI2 (TSI2 group), *L. fermentum* S2 (S2 group), and mixed TSI2 and S2 strains at 1:1 (MIX group), were verified. Six-week-old male Sprague-Dawley rats were divided into five groups and administered the following diets: group NOR, normal diet with oral saline administration; group HF, high-fat diet (HD) with oral saline administration; group TSI, HD and *L. fermentum* TSI-fermented yogurt; group S2, HD and L. fermentum S2-fermented yogurt; and group MIX, HD and MIX-fermented yogurt. After eight weeks, the HD groups displayed significantly increased body weight and fat, serum cholesterol, and abdominal adipose tissue levels. However, serum HDL cholesterol levels were higher, triglyceride levels were lower, and abdominal adipocytes were smaller in the TSI and S2 groups than in the HF group. These results indicate that *L. fermentum* TSI reduces abdominal fat and improves blood lipid metabolism in HD-induced obese rats.

## 1. Introduction

Lactic acid bacteria (LAB) have been widely studied and used worldwide for their health benefits. LAB species are a very important resource in the dairy industry for manufacturing cheese and fermented milk products. Yogurt is a popular dairy product with widespread worldwide consumption. Studies have reported health benefits of yogurt owing to its probiotic content comprising live LAB that prevent pathogen growth, by producing organic acids and bacteriocins in the gastrointestinal tract [1,2]. Yogurt helps reduce blood cholesterol, eliminate endotoxins, and assimilate vitamins and minerals and enhance digestive functions including nutrient and lactose absorption. Additionally, yogurt helps prevent colorectal cancer and diabetes and helps prevent and manage hyperlipidemia [3]. Furthermore, yogurt contains various LAB and stimulates the immune system and stabilizes the gut microflora [4,5]. Consequently, yogurt rich in probiotics has attracted increasing attention [6,7].

Obesity is a serious public health concern, characterized by excessive accumulation of body fat, resulting in an increase in body weight and leading to metabolic and chronic disorders such as diabetes, hypertension, arteriosclerosis, insulin resistance, and cardiovascular diseases [8,9,10]. The World Health Organization (WHO) has reported the environmental, genetic, and neuroendocrine factors associated with obesity, along with infectious agents [11,12]. The bacteria commonly present in the human gut regulate energy and nutrient absorption differently in obese and lean individuals, suggesting that the gut microbiota play an important role in the pathogenesis of obesity [13]. Treatment of obesity with a probiotic-rich diet is a major advancement in the treatment of obesity [13].

In the past decade, Mongolian dairy products have been consumed as not only energy sources but also traditional medicines [14]. Mongolian traditional fermented milk products vary in accordance with their source of milk: *airak* (horse), *hoormog* (camel), *undaa* (cow), and *tarag* (yaks or cattle). Several studies have investigated the microbial content and the diversity of LAB in Mongolian dairy products [15,16,17,18,19]. *L. fermentum* is a heterofermentation LAB belonging to family Lactobacillaceae, phylum Firmicutes. LAB are generally recognized as safe (GRAS) microbes and widely used in the production of fermented food products. *L. fermentum* is considered a potential probiotic organism because of its desirable probiotic characteristics including its beneficial effects on immune function, nonpathogenicity, and being a part of the natural internal environment of humans [20]. However, few studies have focused on the functional and metabolic aspects of yogurt in animal models. Therefore, the potential effects of these products are unknown.

This study aimed to produce yogurt made with strains identified as *Lactobacillus fermentum* TSI2 and *L. fermentum* S2, derived from Mongolian fermented milk, and determine its quality characteristics. Additionally, we investigated its effects on the serological characteristics and morphology of adipocytes in rats with high-fat-diet (HD)-induced obesity.

## 2. Materials and Methods

### 2.1. Microbial Fermentation

*L. fermentum* TSI and *L. fermentum* S2 were isolated from Mongolian traditional fermented milk at the Laboratory of Biotechnology in Konkuk University (Seoul, Korea). These strains were subcultured twice in MRS (Man Rogosa and Sharpe) broth (Oxoid Ltd., Basingstoke, England) at 37 °C for 24 h, and cells were harvested via centrifugation at 2000× *g* for 15 min at 4 °C. The pelleted cells were resuspended in 1 mL of sterilized market milk. The final concentration of cells was adjusted to 8 log CFU mL^−1^ for inoculation into the sterilized milk to obtain yogurt (1%, v/v).

### 2.2. Yogurt Preparation

After mixing market milk (cow), skim milk, pectin, and white sugar (Table 1), the mixture was homogenized for 5 min, using a homogenizer. Thereafter, the homogenate was sterilized for 30 min at 85 °C, cooled to 42 °C, and incubated at pH 4.5. Groups of treatment using different strains were cultured at 37 °C for 18 h until pH reached 4.5.

### 2.3. pH, Titratable Acidity, and LAB Density

pH was measured using a pH meter (pH 900, Precisa Co., Dietikon, Switzerland). Titratable acidity was determined for all groups through neutralization titration up to pH 8.3, using distilled water (3 g of stored yogurt sample in 27 mL of water). Thereafter, 0.1 N NaOH was used to estimate the amount of lactic acid (%) using the following equation:LA% = [(10 × V_NaOH_ × 0.009 × 0.1)/W] × 100%(1)

10 = Dilution factor

W = Weight of sample (g) for titration

V_NaOH_ = Volume of NaOH used to neutralize the lactic acid

= Normality of NaOH.

The LAB density was determined using the streak plate method with MRS agar (Oxoid Ltd., Hampshire, UK). Samples (100 μL each) were serially diluted with 0.85% NaCl solution (900 μL). After spreading and smearing the diluted solution (100 μL) onto MRS agar plates, they were cultured at 37 °C for 24 h. The total number of viable cells was expressed as a log-transformed value.

### 2.4. Viscosity and Syneresis

The viscosity of the yogurt sample stored at 4 °C was determined using a viscometer (Model LVDV-E, Brookfield Engineering Lab. Inc., Middleboro MA, USA) at 50 rpm every minute, using Spindle No. 63 for 5–8 min. Syneresis of yogurt was measured using the method of Keogh and O’Kennedy [21] with modification. Briefly, yogurt (20 g) was centrifuged at 10,000 rpm for 10 min at 4 °C. The difference in weight between the residue and supernatant was then calculated.
Syneresis = weight of supernatant (g)/weight of sample (g) × 100%(2)

### 2.5. Animals and Treatments

Forty 6-week-old male Sprague-Dawley (SD) rats, with an initial body weight of 200 g, were purchased from the Central Laboratory of Animal Inc. (Seoul, Korea). Rats were individually housed in stainless steel cages and fed a standard diet (AIN-93G, Dyets Inc., Bethlehem, PA, USA) for 1 week to stabilize the environmental conditions. Feeding was restricted to 20–25 g, and water was supplied ad libitum. The rats were exposed to a 12:12-h light/dark cycle and a constant temperature of 25 ± 1 °C and humidity of 55 ± 5% during the experimental period. After 1 week of acclimation, the rats were randomly divided into 5 groups with no significant difference in body weight in each group: normal diet control fed with AIN-93G (NOR, *n* = 8), HD-fed control (HF, *n* = 8), HD-fed rats orally administered yogurt fermented by *L. fermentum* TSI (TSI, *n* = 8), HD-fed rats orally administered yogurt fermented by *L. fermentum* S2 (S2, *n* = 8), HD-fed rats orally administered yogurt fermented by a mixture of *L. fermentum* TSI and S2 (MIX, *n* = 8). CON in yogurt was excluded from animal testing to minimize the number of rats to be euthanized, owing to the absence of significant differences in quality characteristics in comparison with the treatment groups. NOR and HF groups were orally treated with 0.95% saline solution 1 mL 200 g^−1^ body weight. TSI, S2, and MIX groups were each administered 1 mL yogurt per 200 g body weight. Compositions of the normal and 45% HD diets are presented in Table 2. Food intake was measured daily and body weight was measured weekly. Animal experiments were performed in accordance with the guidelines of the Animal Use Committee of Konkuk University (approved No. KU17078).

### 2.6. Analysis of Blood Serum and Organ Weight

After the 8-week experimental period, all rats were fasted 12 h before euthanasia. Blood samples from the abdominal artery of each rat were collected into tubes under light anesthesia in an ether-saturated chamber. Serum was separated through centrifugation at 3000× *g* for 15 min at 4 °C. Serum was analyzed for total cholesterol (TC), high density lipoprotein cholesterol (HDL-C), low density lipoprotein cholesterol (LDL-C), triglyceride, glucose, leptin, adiponectin, aspartate transaminase (AST), and alanine transaminase (ALT) levels. The liver, kidney, spleen, abdominal fat, and epididymal fat were dissected out and weighed. The gut flora were assessed in the intestine.

### 2.7. Measurement of Adipocytes

Abdominal and epididymal fat tissue were dissected out and rapidly fixed with 10% formaldehyde solution. The samples were filtered with a 250-µm nylon filter to eliminate fibrous tissue and intact fixed adipose tissue, as previously described [22]. Tissue was shredded with phosphate buffer saline (PBS) and filtered through a 25-µm nylon filter to trap the fixed adipose tissue, and the cells were washed sufficiently with PBS. Using the method of van Goor et al. [23], cell suspensions were placed on a glass slide and subjected to hematoxylin and eosin (H&E) staining and then visualized using a microscope (Olympus IX71, Olympus Co., Tokyo, Japan), and the adipocyte sizes were measured using ImageJ software program (NIH, Bethesda, Rockville, MD, USA).

### 2.8. Statistical Analysis

All data from triplicate experiments are expressed as mean ± standard deviation (SD) values and were analyzed using one-way analysis of variance using SPSS/PC Statistics 18.0 software (SPSS Inc., Chicago, IL, USA), followed by Tukey’s test for post hoc analysis. Values of *p* < 0.05 were considered statistically significant.

## 3. Results and Discussion

### 3.1. pH, Titratable Acidity, and the Number of LAB

The pH changed before and after fermentation from 6.60 to 4.45 for CON, 6.62 to 4.48 for TSI2, 6.64 to 4.53 for S2, and 6.66 to 4.56 for MIX (for all, *p* < 0.05). Upon stabilization for 1 d, the pH of CON was low at 4.17, 4.29 for TSI2, 4.28 for S2, and 4.26 for MIX. The pH did not significantly change among groups (*p* > 0.05) (Table 3).

Lactose fermentation decreased the pH, which may be because the conversion of lactose to lactic acid is increased by the metabolic activity of bacteria [24]. According to Lee and Hwang (2006) [25], the optimum pH of commercially available fermented milk is 3.27–4.59. Herein, the pH of yogurt after 24-h stabilization was within this range, suggesting no differences in quality between yogurt produced herein and commercially available fermented milk.

The value of titratable acidity after 24-h stabilization peaked at 1.01% for CON, and it was 0.91% for TSI2, 0.91% for S2, and 0.90% for MIX; these values did not significantly differ (*p* > 0.05) (Table 4). Titratable acidity is influenced by the level of nonfat solid substances such as citrates, proteins, and phosphates [26]. Davis (1970) reported that the titratable acidity of commercially available fermented milk was 0.72%–1.20% [27]. The titratable acidity of yogurt produced herein was also within this normal range.

The LAB density was 10.07 log CFU/g in MIX, 9.43 log CFU/g in CON, 9.40 log CFU/g in TSI2, and 9.28 log CFU/g in S2 (Table 4). Herein, no significant differences in the viable LAB count were observed among all groups (*p* > 0.05). According to Codex, the total number of *Lactobacillus* in fermented beverages should exceed 7 log CFU/g. Herein, the LAB density exceeded 7 log CFU/g in all groups of yogurt. According to Kroger and Weaver (1973) [28], if yogurt is produced with probiotics with a mixed culture (more than two LAB), the acidity and viable cell count are greater than those of monocultures.

### 3.2. Viscosity and Syneresis

The viscosity and syneresis of yogurt produced using CON, TSI2, S2, and MIX are summarized in Table 5. The viscosity of CON produced with commercial starter peaked at 827 cp. The viscosity of TSI2 was 421 cp; S2, 378 cp; MIX, 317 cp (being the lowest) (*p* > 0.05).

CON displayed the lowest syneresis. For yogurt produced with isolated strains, no significant difference in syneresis was observed. Syneresis decreased in the order of TSI2 > S2 > MIX. Syneresis was inversely associated with pH. At a low pH, e.g., <pH 5.0, casein approaches its isoelectric point. Accordingly, electrostatic repulsion is minimized owing to interactions among proteins [29,30]. Since proteins interact with water, the moisture bolding potential of the protein matrix is decreased under pH 5.0 [31]. Syneresis separation is easily achieved because the structure of milk proteins is affected at low pH [32].

### 3.3. Body Weight and Food Intake

The changes in body weight, food intake, and food efficiency over eight weeks are summarized in Table 6. After the experiment, the final body weight of all the groups treated with 45% HD over eight weeks was significantly increased greater than that of the NOR group (*p* < 0.05). No significant differences were observed among the HD groups. Furthermore, no significant difference in food intake was observed among all groups because of equal feeding restrictions. However, the food efficiency ratio (FER) was significantly higher in the HD groups than in the NOR group (*p* < 0.05). As shown in Figure 1, the body weight of rats in all the groups increased gradually every week, especially in the HF group more than in the NOR group. Differences in body weight among groups were remarkable at week 2 and significant after week 7 of the treatment. At the end of the experiment, no significant difference was observed among the treatment groups except in the NOR group.

NOR, normal diet control fed with AIN-93G and treated with oral saline; HF, 45% high-fat diet-fed control treated with oral saline; TSI, 45% high-fat diet fed and orally treated with yogurt fermented by *Lactobacillus fermentum* TSI; S2, 45% high-fat diet fed and orally treated with yogurt fermented by *L. fermentum* S2; MIX, 45% high-fat diet fed and orally treated with yogurt fermented by *L. fermentum* TSI and *L. fermentum* S2 mixed culture.

### 3.4. Weight of Organs and Fat

Weights of organs and fat pads are listed in Table 7. The liver, kidney, spleen, abdominal fat, and epididymal fat pad were weighed, and no significant differences were observed among the HD groups. HD induced the accumulation of abdominal and epididymal fat in comparison with the normal diet. Similar results were observed for body weight herein. However, the weight of epididymal fat in the S2 group was significantly lower than in other HD groups, suggesting that yogurt fermented with *L. fermentum* S2 might prevent epididymal fat accumulation in HD rats. Yogurt fermented by *L. plantarum* Q180 reduced epididymal fat levels in diet-induced obese rats, concurrent with a previous report [33].

### 3.5. Serum Biochemistry

Serum cholesterol, triglyceride, glucose, leptin, AST, and ALT levels are summarized in Table 8. Compared with the HF group, the TSI group displayed a significantly increased HDL-cholesterol level (*p* < 0.05), which was slightly but not significantly higher in the S2 group. Serum triglyceride levels were slightly but not significantly lower in the TSI group than in the HF group (*p* > 0.05). Adiponectin levels were significantly higher in the TSI group than in the HF group and similar to that in the NOR group (*p* < 0.05). These results indicate that feeding obese rats with yogurt fermented by *L. fermentum* TSI increased adiponectin levels. Adiponectin regulates numerous metabolic processes including regulation of energy balance, glucose homeostasis, and lipid metabolism, and obesity is associated with decreased adiponectin levels. The exact regulatory mechanism remains unknown, although adiponectin may be regulated by post-translational mechanisms in cells [34].

### 3.6. Adipocyte Tissue Size

The abdominal fat pad was rapidly separated and subjected to H&E staining. Adipocyte size in the abdominal fat pad was microscopically determined (Figure 2).

A, NOR, normal diet control fed with AIN-93G and treated with oral saline; B, HF, 45% high-fat diet-fed control treated with oral saline; C, TSI, 45% high-fat diet fed and orally treated with yogurt fermented by *Lactobacillus fermentum* TSI; D, S2, 45% high-fat diet fed and orally treated yogurt fermented by *L. fermentum* S2; E, MIX, 45% high-fat diet fed and orally treated with yogurt fermented by *L. fermentum* TSI and *L. fermentum* S2 mixed culture.

The adipocyte size of HD-fed groups varied widely in comparison with that of the NOR group at the same magnification level. Rats fed with fermented yogurt had smaller adipocytes than those in the HF group. Concurrently, numerous studies have reported that adipocyte size is increased during obesity [35,36]. Each cell was imaged ten times and the average adipocyte size was measured using ImageJ software (Figure 3). 

NOR, normal diet control fed with AIN-93G and treated with oral saline; HF, 45% high-fat diet-fed control treated with oral saline; TSI, 45% high-fat diet fed and orally treated with yogurt fermented by *Lactobacillus fermentum* TSI; S2, 45% high-fat diet fed and orally treated with yogurt fermented by *L. fermentum* S2; MIX, 45% high-fat diet fed and orally treated with yogurt fermented by *L. fermentum* TSI and *L. fermentum* S2 mixed culture.

HD-fed groups presented a significantly greater adipocyte size than those administered a normal diet (*p* < 0.05). Accordingly, the TSI and MIX groups presented significantly smaller adipocytes than the HF group. Adipocyte tissue growth occurs via hyperplasia and hypertrophy. Furthermore, a reduction in the adipocyte size prevents hyperplasia and hypertrophy because of an increase in adipocyte-stimulating factors triggering adipogenesis via preadipocyte differentiation [37].

The quality characteristics of yogurt fermented by LAB species derived from a Mongolian fermented milk product including *L. fermentum* TSI and *L. fermentum* S2 were found to be suitable for commercialization. The fermentation was confirmed to be well-processed because the pH values were within the optimum pH range of fermented milk on the market. In all groups, the number of LAB was over 7 log CFU/g, thus meeting the minimum requirement for fermented food products. The yogurt prevented obesity by decreasing the adipocyte size in the abdominal fat tissue of HD-induced obese rats. In particular, yogurt obtained from *L. fermentum* TSI decreased the abdominal fat and improved serum HDL-cholesterol levels and adiponectin secretion. However, the synergistic effect of the combination of two strains remains unclear. Therefore, dairy products fermented by *L. fermentum* TSI, isolated from Mongolian traditional dairy products, may be a functional probiotic in improving blood cholesterol and body fat accumulation.

## Figures and Tables

**Figure 1 foods-09-00594-f001:**
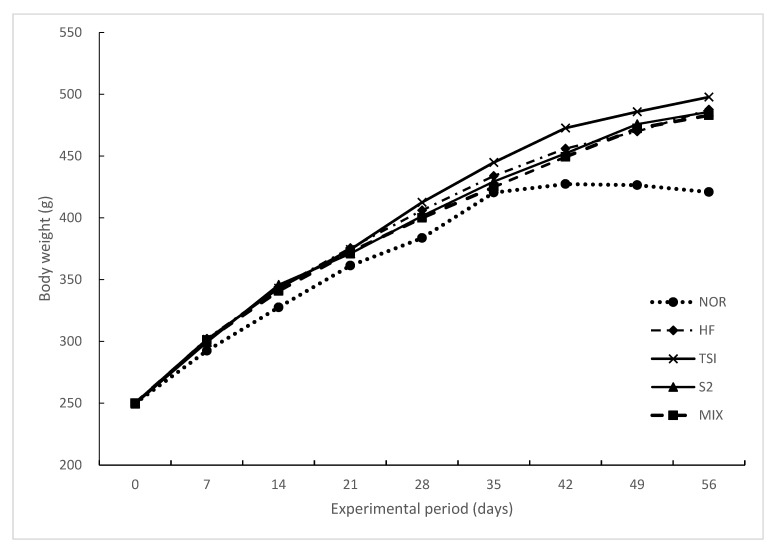
Changes in the body weight gain of rats upon oral treatment for 8 weeks.

**Figure 2 foods-09-00594-f002:**
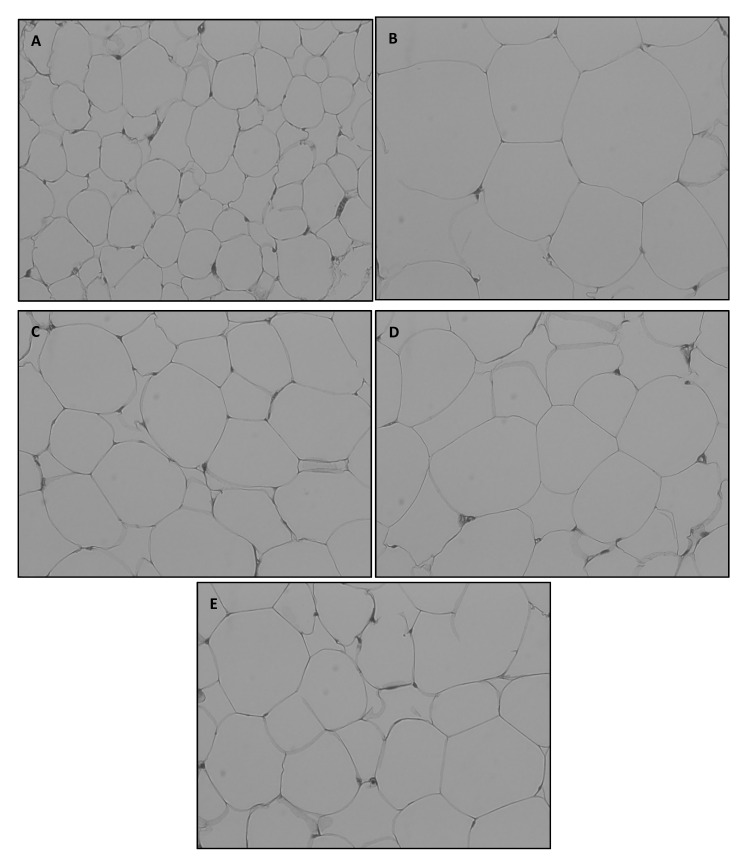
Representative hematoxylin and eosin staining of omental adipose tissue (OMAT) in SD rats exposed to experimental diet for 8 weeks (original magnification, × 200).

**Figure 3 foods-09-00594-f003:**
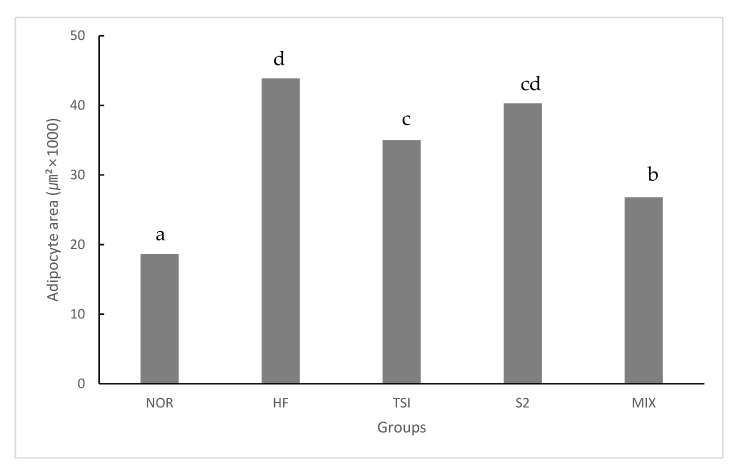
Adipocyte size in abnormal adipose tissue in SD rats on an experimental diet for 8 weeks.

**Table 1 foods-09-00594-t001:** Yogurt components.

Ingredients (g)	CON	Treatments
Milk	850	850
Powdered skim milk	40	40
Sugar	15	15
Pectin	2	2
Distilled water	105	105
Starter	0.2 (g)	10 (mL)
Total	1012.2	1022

CON: control; Treatments: yogurt added with 1% Lactic acid bacteria.

**Table 2 foods-09-00594-t002:** Nutritional composition of experimental normal diet (AIN-93G) and 45% high-fat diet.

Ingredient ^1)^	Standard Diet (AIN-93G Purified Diet)		High-Fat Diet (Rodent Diet with 45% kcal% Fat)	
	gm%	kcal%	gm%	kcal%
Protein	20.0	20.3	24.0	20.0
Carbohydrate	64.0	63.9	41.0	35.0
Fat	7.0	15.8	24.0	45.0
Total kcal/gm	3.9		4.7	
Casein, 30 Mesh	200	800	233.1	932
L-cystine	3	12	3.5	14
Corn Starch	397	1590	84.8	339
Maltodextrin	132	528	116.5	466
Sucrose	100	400	201.4	805
Cellulose	50	0	58.3	0
Soybean Oil	60	630	29.1	262
Lard ^2)^	0	0	206.8	1862
t-Butylhydroquinone	0.014	0	0	0
Mineral Mix	35	0	11.7	0
Dicalcium Phosphate	0	0	15.1	0
Calcium carbonate	0	0	6.4	0
Potassium citrate, 1H_2_O	0	0	19.2	0
Vitamin Mix	10	40	11.7	47
Choline Bitartrate	2.5	0	2.3	0
FD&C Red Dye #40	0	0	0.1	0
Total	1000	4000	1000	4727.6

^1)^ Formula Product # D10012G (standard diet), #D12451 (high-fat diet)-added Mineral mix S10022G, S10026 and Vitamin mix V10037, V10001 were used, respectively. (Research diet, Inc., New Brunswick, NJ, USA). ^2)^ Typical analysis of cholesterol in lard = 0.72 mg/gram.

**Table 3 foods-09-00594-t003:** The change of pH of each starter.

	CON	TSI2	S2	MIX
Before fermentation pH	6.60 ± 0.01 ^c^	6.62 ± 0.01 ^bc^	6.64 ± 0.03 ^ab^	6.66 ± 0.02 ^a^
Immediately after fermentation pH	4.45 ± 0.01 ^ns^	4.48 ± 0.01	4.53 ± 0.13	4.56 ± 0.11
After stabilizing pH	4.17 ± 0.02 ^b^	4.29 ± 0.02 ^a^	4.28 ± 0.03 ^a^	4.26 ± 0.01 ^a^

Different letter superscripts within the same line indicate significant differences, *p* < 0.05. CON: control; TSI2: Yogurt produced from identified cow milk; S2: Yogurt produced from identified donkey milk; MIX: Yogurt prepared from mixed identified cow milk and donkey milk (1:1). All values are means ± standard deviation for triplicate.

**Table 4 foods-09-00594-t004:** Titratable acidity and lactic acid bacteria of each starter after 24-h stabilization.

	CON	TSI2	S2	MIX
Titratable acidity (%)	1.01 ± 0.00 ^a^	0.91 ± 0.04 ^b^	0.88 ± 0.04 ^b^	0.90 ± 0.01 ^b^
Lactic acid bacteria (Log CFU/g)	9.43 ± 0.33 ^ns^	9.40 ± 0.34	9.28 ± 0.27	10.07 ± 0.53

Different letter superscripts within the same line indicate significant differences, *p* < 0.05. CON: control; TSI2: Yogurt produced from identified cow milk; S2: Yogurt produced from identified donkey milk; MIX: Yogurt produced from mixed identified cow milk and donkey milk (1:1). All values are means ± standard deviation for triplicate.

**Table 5 foods-09-00594-t005:** Viscosity and syneresis of each starter after 24-h stabilization.

	CON	TSI2	S2	MIX
Viscosity (cp)	827.00 ± 24.52 ^a^	420.75 ± 94.24 ^b^	377.50 ± 49.18 ^b^	316.51 ± 44.37 ^b^
Syneresis (%)	58.00 ± 0.58 ^b^	69.43 ± 0.37 ^a^	71.33 ± 2.54 ^a^	73.35 ± 1.97 ^a^

Different letter superscripts within the same line indicate significant differences, *p* < 0.05. CON: control; TSI2: Yogurt produced from identified cow milk; S2: Yogurt produced from identified donkey milk; MIX: Yogurt produced from mixed identified cow milk and donkey milk (1:1). All values are means ± standard deviation for triplicate.

**Table 6 foods-09-00594-t006:** Body weight changes, food intakes, and food efficiency ratio of Sprague-Dawley (SD) rats during the experimental period (*n* = 6).

	NOR ^1)^	HF	TSI	S2	MIX
	**Body Weight (g/rat)**				
Initial	249.8 ± 7.0 ^NS^	249.8 ± 6.3	250.2 ± 6.0	249.5 ± 6.1	250.0 ± 6.3
Final	420.8 ± 4.1 ^a^	487.5 ± 16.2 ^b^	497.7 ± 26.9 ^b^	485.7 ± 29.8 ^b^	483.0 ± 15.3 ^b^
Weight gain	153.0 ± 3.52	237.7 ± 10.1	247.5 ± 21.1	236.2 ± 24.7	233.0 ± 9.3
Weight gain rate (%) ^2)^	0	55.3	61.8	54.4	52.3
Weight loss rate (%) ^3)^	55.3	0	−4.0	0.6	2.0
	**Food Intake**				
Total (g/rat)	1008.0 ± 0.0 ^NS^	1009.2 ± 99.1	1009.8 ± 72.1	1007.0 ± 40.8	1008.5 ± 75.9
Daily (g/daily/rat)	18.0 ± 0.0 ^NS^	18.0 ± 1.5	18.0 ± 1.1	18.0 ± 0.6	17.9 ± 1.2
FER ^4)^	15.2 ± 0.3 ^a^	23.7 ± 1.4 ^b^	24.5 ± 0.7 ^b^	23.4 ± 1.8 ^b^	23.2 ± 1.0 ^b^

Different letter superscripts within the same line indicate significant differences, *p* < 0.05. ^1)^ NOR, normal diet control fed with AIN-93G and treated with oral saline; HF, 45% high-fat diet fed control treated with oral saline; TSI, 45% high-fat diet-fed and orally treated yogurt fermented by *Lactobacillus fermentum* TSI; S2, 45% high-fat diet-fed rats orally treated with yogurt fermented by *Lactobacillus fermentum* S2; MIX, 45% high-fat diet fed and orally treated with yogurt fermented by *Lactobacillus fermentum* TSI and *Lactobacillus fermentum* S2 mixed culture. ^2)^ Weight gain rate compared with NOR group, ^3)^ Weight loss rate compared with HF group, ^4)^FER: Food efficiency ratio, Wt. gain (g)/Food intake (g) × 100.

**Table 7 foods-09-00594-t007:** Organ weight (g) and fat weight of SD rats fed with experimental diet for 8 weeks (*n* = 6).

	NOR	HF	TSI	S2	MIX
Liver	8.93 ± 0.12 ^a^	13.29 ± 1.98 ^bc^	12.23 ± 1.01 ^bc^	11.78 ± 0.69 ^b^	13.70 ± 0.48 ^c^
Kidney	2.27 ± 0.06 ^a^	2.55 ± 0.15 ^b^	2.49 ± 0.21 ^ab^	2.39 ± 0.11 ^ab^	2.48 ± 0.13 ^ab^
Spleen	0.71 ± 0.03 ^b^	0.66 ± 0.06 ^ab^	0.62 ± 0.05 ^ab^	0.62 ± 0.06 ^a^	0.68 ± 0.07 ^ab^
Abdominal fat	9.71 ± 1.85 ^a^	18.36 ± 3.10 ^b^	16.71 ± 1.71 ^b^	16.96 ± 3.26 ^b^	18.53 ± 3.87 ^b^
Epididymal fat	8.21 ± 1.83 ^a^	13.42 ± 4.34 ^b^	14.38 ± 1.95 ^b^	13.11 ± 3.60 ^ab^	16.96 ± 2.28 ^b^

Different superscripts within the same line indicate significant differences, *p* < 0.05. NOR, normal diet control fed with AIN-93G and treated with oral saline; HF, 45% high-fat diet-fed control treated with oral saline; TSI, 45% high-fat diet-fed rats treated orally with yogurt fermented by *Lactobacillus fermentum* TSI; S2, 45% high-fat diet-fed and orally treated with yogurt fermented by *Lactobacillus fermentum* S2; MIX, 45% high-fat diet fed and orally treated with yogurt fermented by *Lactobacillus fermentum* TSI and *Lactobacillus fermentum* S2 mixed culture.

**Table 8 foods-09-00594-t008:** Blood serum levels of cholesterol, triglyceride, glucose, leptin, aspartate transaminase, and alanine transaminase in SD rats fed with experimental diet for 8 weeks.

	NOR	HF	TSI	S2	MIX
Total cholesterol	57.2 ± 6.68 ^a^	72.8 ± 10.85 ^ab^	70.3 ± 20.72 ^ab^	81.7 ± 9.83 ^b^	76.3 ± 5.85 ^ab^
HDL-cholesterol	55.5 ± 8.07 ^a^	56.2 ± 5.16 ^ab^	78.7 ± 12.48 ^c^	71.2 ± 11.99 ^bc^	63.7 ± 1.86 ^abc^
LDL-cholesterol	11.0 ± 2.37 ^ab^	9.5 ± 0.55 ^a^	14.5 ± 4.04 ^b^	10.0 ± 2.61 ^a^	10.8 ± 2.32 ^ab^
Triglyceride	75.8 ± 8.91 ^a^	125.0 ± 58.85 ^ab^	103.7 ± 12.80 ^ab^	130.2 ± 35.07 ^ab^	160.7 ± 65.55 ^b^
Glucose	70.0 ± 11.05 ^a^	129.5 ± 14.39 ^b^	125.7 ± 19.37 ^b^	154.8 ± 10.55 ^c^	157.5 ± 12.31 ^c^
Leptin	1304.5 ± 219.75 ^a^	3921.6 ± 937.88 ^b^	3453.7 ± 1119.57 ^b^	3964.3 ± 1779.8 ^b^	4132.8 ± 432.31 ^b^
Adiponectin	19,974.9 ± 3276.02 ^c^	12033.7 ± 2219.51 ^a^	17447.2 ± 3840.83 ^bc^	12587.7 ± 1969.12 ^a^	12,923.3 ± 2301.64 ^ab^
AST	169.5 ± 13.74 ^c^	99.0 ± 22.86 ^ab^	123.8 ± 23.89 ^bc^	99.0 ± 22.16 ^ab^	78.0 ± 16.88 ^a^
ALT	35.7 ± 7.75 ^b^	34.2 ± 4.4 ^b^	32.2 ± 3.19 ^ab^	25.5 ± 4.23 ^a^	27.8 ± 3.71 ^ab^

Different superscripts within the same line indicate significant differences, *p* < 0.05. NOR, normal diet control fed with AIN-93G and treated with oral saline; HF, 45% high-fat diet-fed control treated with oral saline; TSI, 45% high-fat diet fed and orally treated with yogurt fermented by *Lactobacillus fermentum* TSI; S2, 45% high-fat diet fed and orally treated with yogurt fermented by *Lactobacillus fermentum* S2; MIX, 45% high-fat diet fed and orally treated with yogurt fermented by *Lactobacillus fermentum* TSI and *Lactobacillus fermentum* S2 mixed culture.

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
