# Peer review of "Effect of Yogurt Fermented by Lactobacillus Fermentum TSI and L. Fermentum S2 Derived from a Mongolian Traditional Dairy Product on Rats with High-Fat-Diet-Induced Obesity"

_foods, 2020, doi:10.3390/foods9050594_

Round 1

Reviewer 1 Report

Dear authors, this is a very well-written paper and of considerable interest to many. I just have a couple of very minor comments which could be addressed for clarity:

On Line 41 - change the word 'furthermore' to 'additionally' or a similar word. Furthermore appears in the previous sentence.

Can you clarify for the reader what Mongolian traditional fermented milk is? For example, is it bovine milk, how is it fermented? is it a commercial product or an artisan product etc...

On Line 79 - do you mean that the samples were incubated to pH 4.5? I think that's what you mean - as it is written it reads as if you changed the pH to 4.5 before incubation

When you did the statistical analysis on the viscosity values, did you include the control in the data set? I'm just wondering because, if it was included, then I'm guessing that there should be a statistically significant difference between the viscosities?

Overall, an excellent manuscript

Author Response

  1. On Line 41 - change the word 'furthermore' to 'additionally' or a similar word. Furthermore appears in the previous sentence.

→We changed the the word as you said.

  1. Can you clarify for the reader what Mongolian traditional fermented milk is? For example, is it bovine milk, how is it fermented? is it a commercial product or an artisan product etc...

→We isolated from Mongolian. TSI2 and S2 is not commercial strain but isolates.

  1. On Line 79 - do you mean that the samples were incubated to pH 4.5? I think that's what you mean - as it is written it reads as if you changed the pH to 4.5 before incubation

→ I changed the sentence.

  1. When you did the statistical analysis on the viscosity values, did you include the control in the data set? I'm just wondering because, if it was included, then I'm guessing that there should be a statistically significant difference between the viscosities?

→ Since CON was included, it was confirmed that CON was a and the experimental group was b.

Reviewer 2 Report

Effect of Yogurt Fermented by Lactobacillus fermentum TSI and L. fermentum S2 Derived from a Mongolian Traditional Dairy Product on Rats with High-Fat-Diet-Induced Obesity

General comments

The topic of the manuscript is very interesting, particularly considering that data about fermented products and its potential effect on rats with very interesting FIG 2. Generally speaking, the article is well written, but I have some questions that need to be answered or explained especially to first part.

Abstract

Line 54-56

Mongolian traditional fermented milk products vary in accordance with their source of milk: airak (horse), hoormog (camel), undaa (cow), and tarag (yaks or 56 cattle).

As it is written this traditional fermented milk can be produced from different kind of animals milk. That’s why my next questions

Line 65 derived from Mongolian fermented milk

What source of milk? Mare’s, camel’s?

If it is mare’s milk have authors information about it? Lactation period, age of mares?

How was it collected? How was it fermented traditionally means spontaneous?

Was the fermented milk in bottles or was it derived strictly from farmers?

Materials and Methods

Line 74 sterilized market milk

What is the fat content in this milk?

Line 77-78  

After mixing milk- it is no information what kind of milk (cow’s? donkey’s, mares?)

In my opinion

After mixing milk, skim milk, pectin, and white sugar (the reference to Table 1 should be here), the mixture was homogenized for 5 min, 78 using a homogenizer.

Line 128 

After the 8-week experimental period

Why such a period, not shorter or longer? Do authors have any experience with this?

In my opinion it is necessary to add points

  1. Discussion
  2. Conclusion

Author Response

  1. Line 65 derived from Mongolian fermented milk
  • What source of milk? Mare’s, camel’s?

→ TSI2 is cow milk, and S2 is Mare milk.

  • If it is mare’s milk have authors information about it? Lactation period, age of mares?

→We obtained the Mongolian yogurt from nomads.

  • How was it collected? How was it fermented traditionally means spontaneous?

→Mongolian said that milk was made by Mongolian’s nomads.

  • Was the fermented milk in bottles or was it derived strictly from farmers?

→Nomads took the milk and gave it to us.

  1. Line 74 sterilized market milk
  • What is the fat content in this milk?

→The fat content of market milk is 3.4%.

  1. Line 77-78  
  • After mixing milk- it is no information what kind of milk (cow’s? donkey’s, mares?)

→In this sentence, milk refers to cow milk. Since it can cause confusion, the text was modified to be market milk (cow).

  • In my opinion

After mixing milk, skim milk, pectin, and white sugar (the reference to Table 1 should be here), the mixture was homogenized for 5 min, 78 using a homogenizer.

→I changed the position.

Line 128 

After the 8-week experimental period

Why such a period, not shorter or longer? Do authors have any experience with this?

In my opinion it is necessary to add points

→In general, the appropriate drug administration period is 4 weeks or more, and anti-obesity effects are not observed after 6 weeks of administration, and there is a weight loss effect when administered for 8 weeks or more. Therefore, most of the papers related to anti-obesity effects are confirmed until 8 weeks. Below, we add the papers that had been conducted for up to 8 weeks.

√ Yeon, S.J.; Kim, S.K.; Kim, J.M.; Lee, C.H. Effects of Fermented Pepper Powder on Body Fat Accumulation in Mice Fed a High-Fat Diet. Journal Bioscience, Biotechnology, and Biochemistry. 2013, 11(77).

√ Zhao, L.; Zhang, Q.; Ma, W.; Tian, W.; Shen, H.; Zhou, M. A combination of quercetin and resveratrol reduces obesity in high-fat diet-fed rats by modulation of gut microbiota. Food & Function. 2017, 8.

√ Raman, H.A.; Sahib, N.G.; Saari, N.; Abas, F.; Ismail, A.; Mumtaz, M.W.; Hamid, A.A. Anti-obesity effect of ethanolic extract from Cosmos caudatus Kunth leaf in lean rats fed a high fat diet. BMC Complementary Medicine and Therapies. 2017, 17.

  1. Discussion

→We have added more discussion.

  1. Conclusion

→We have added more conclusion.